# OpenReview forum: "Look Before You Leap: Universal Emergent Mechanism for Retrieval in Language Models"
_ICLR.cc/2025/Conference — ICLR 2025 Poster_

### Official Review · Reviewer_diV7 · 2024-10-21

**Soundness:** 4
**Presentation:** 3
**Contribution:** 3
**Rating:** 6
**Confidence:** 4

**Summary:**

This paper investigates how language models (LMs) solve retrieval tasks across diverse situations. The authors introduce ORION, a collection of structured retrieval tasks spanning six domains, from text understanding to coding. They apply causal analysis on ORION for 18 open-source language models ranging from 125 million to 70 billion parameters. The study reveals that LMs internally decompose retrieval tasks in a modular way: middle layers at the last token position process the request, while late layers retrieve the correct entity from the context. The authors demonstrate a proof-of-concept application for scalable internal oversight of LMs to mitigate prompt injection while requiring human supervision on only a single input.

**Strengths:**

1. The introduction of ORION provides a valuable resource for studying LM behavior across a wide range of retrieval tasks. By creating a structured dataset with consistent abstract representations, the authors enable systematic comparisons across different models and task types. This data-centric approach is particularly useful as it allows for scalable analysis of LM behavior.

2. The discovery of a universal emergent mechanism for retrieval tasks across different model sizes and architectures is a decent contribution. The authors' finding that middle layers process the request while late layers perform retrieval suggests a fundamental organizing principle in LMs that was previously unknown. This insight could have far-reaching implications for our understanding of LM capabilities and limitations.

3. The paper demonstrates a compelling link between theoretical understanding and practical application. By leveraging their insights into the internal workings of LMs, the authors develop a proof-of-concept for mitigating prompt injection attacks. This application showcases the potential real-world impact of mechanistic interpretability research.

4. The study's comprehensive scope, covering 18 different models across various scales, adds robustness to its findings. This breadth allows the authors to identify patterns that persist across different architectures and training regimes, strengthening the generalizability of their conclusions.

**Weaknesses:**

1. While the paper identifies the modular decomposition of retrieval tasks, it lacks a deep exploration of why this decomposition emerges. The authors could strengthen their analysis by investigating how this modularity relates to the training process or architectural choices in LMs. For instance, they could examine whether the observed layerwise specialization correlates with specific attention patterns or activation distributions.

2. The study focuses primarily on retrieval tasks with single-attribute requests. This limitation raises questions about how well the findings generalize to more complex retrieval scenarios involving multiple attributes or nested queries. Expanding the analysis to include such tasks could provide a more comprehensive understanding of LM retrieval capabilities.

3. The paper's case study on Pythia-2.8b reveals inconsistencies between macro-level modularity and micro-level component behavior. This discrepancy is not fully resolved, leaving open questions about how emergent behaviors arise from the interactions of individual components. A more in-depth analysis of this phenomenon could provide valuable insights into the nature of computation in LMs.

4. The authors' proof-of-concept for mitigating prompt injection is limited in scope. The experiment focuses on a narrow range of distractor types and only two model sizes. A more comprehensive evaluation, including a wider variety of adversarial inputs and model architectures, would strengthen the claims about the method's effectiveness and generalizability.

5. The paper does not adequately address the potential limitations of the residual stream patching technique. For instance, it's unclear how sensitive the results are to the choice of patching layers or whether the observed effects could be artifacts of the intervention method itself. A more thorough discussion of these methodological considerations would enhance the robustness of the study's conclusions.

**Questions:**

1. How does the emergent modularity in retrieval task processing relate to the pre-training objectives or architectures of the studied LMs? Are there specific design choices that might promote or hinder the development of this modular structure?

2. Your analysis focuses on single-attribute retrieval tasks. How do you expect the observed mechanisms to generalize to more complex queries involving multiple attributes or nested retrieval operations? Have you conducted any preliminary investigations in this direction?

3. The case study on Pythia-2.8b reveals that individual components behave differently depending on superficial input features. How do you reconcile this micro-level variability with the macro-level modularity observed across tasks and models? What implications does this have for our understanding of emergent behaviors in LMs?

4. Your proof-of-concept for mitigating prompt injection shows promising results. Have you explored how this technique might scale to more diverse types of adversarial inputs or larger language models? What challenges do you anticipate in applying this method to real-world deployment scenarios?

---

> ### Author Response · Authors · 2024-11-17
> **Influence of architecture and training; and extensions to more advanced queries**
>
> 1.
>
> > How does the emergent modularity in retrieval task processing relate to the pre-training objectives or architectures of the studied LMs? Are there specific design choices that might promote or hinder the development of this modular structure?
>
> The emergent modularity in retrieval task processing likely arises from two key factors:
> 1. **Prevalence in pre-training data**: As demonstrated by the diversity of tasks in ORION spanning question-answering, translation, factual recall, variable binding, induction and code understanding, we can expect next-token prediction requiring retrieval-style abilities to be frequent in pre-training data. Through exposure to many such examples during training, models likely converge towards this modular structure of request processing followed by retrieval, similar to how other emergent capabilities like induction heads develop.
> 2. **Architectural constraints of Transformers**: The sequential nature of the Transformer architecture, where each layer builds on representations from previous layers, creates natural constraints on information processing. While attention mechanisms allow looking at all positions, early layers can only perform relatively simple operations since they work with token-level representations rather than higher-level semantic features. This architectural constraint may naturally push the model towards processing the request starting in the early layers, until the middle layers, where sufficient semantic understanding has developed. Later layers build on this higher-level representation to perform the more complex task of retrieving the correct information from the context.
>
> An important direction for future research, would be to investigate how this modular pattern emerges during training. We complemented the conclusion to include this future direction.
>
> 2.
> > Your analysis focuses on single-attribute retrieval tasks. How do you expect the observed mechanisms to generalize to more complex queries involving multiple attributes or nested retrieval operations? Have you conducted any preliminary investigations in this direction?
>
> One straightforward extension to include multi-attribute question, would be to include prompts of the form:
> “Bob is living in Paris and 31 years old.
> Alice is 50 years old and lives in London.
> Question: what is the age of Bob?”
>
>
> We hypothesis that the internal computation at the last position could be split at two points: a point in early layers where one patch to swap the entity (i.e. Bob / Alice), and later layers where patching changes the attribute (age / city). Such an experiment could also reveal that other attributes are used, for instance maybe the fact that ‘Bob’ appears first might turn out to be relevant. It could also reveal that the two attributes are computed in parallel such that there could be a layer at which we can perform request-patching, but no intermediate points to swap attributes independently.
>
>
> For nested request, we could study questions of the form:
> “[Story]
> What is the name of the cat from the character living in London?”
>
>
> Similar to the case of multiple attributes, we hypothesize that we could split the model at different points to isolate different features of the request: for instance two points at which patching could swap the city of the character or the animal whose name is asked.
> However, we expect that only the largest LM will be able to solve such tasks in a single next-token prediction. Another approach to study more complex queries would be to rely on chain of thought prompting to divide the nested query into elementary queries, and study them in isolation.
> We have not conducted experiments in this direction yet.

---

> ### Author Response · Authors · 2024-11-17
> **Reconciling micro and macro scale observations and a plan to extend the proof of concept to realistic deployment cases**
>
> 3.
> > The case study on Pythia-2.8b reveals that individual components behave differently depending on superficial input features. How do you reconcile this micro-level variability with the macro-level modularity observed across tasks and models? What implications does this have for our understanding of emergent behaviors in LMs?
>
> The apparent tension between micro-level variability and macro-level modularity can be reconciled by understanding how specialized components aggregate to perform general functions. Our case study on Pythia-2.8b revealed that individual attention heads at late layers are specialized to retrieve specific city names - for instance, a head is only activated when retrieving "Cusco" while remaining inactive for other city names. If for every city name, there exists one head at late layer performing the retrieval function, then you can treat this layer as perfoming an abstract 'retrieval' irrespective of the city name.The macro-level modularity emerges from the collective behavior of these specialized components.
>
>
> This relationship between micro and macro behaviors has important implications for our understanding of emergent behaviors in LMs. First, it suggests that emergent behaviors need to be studied at multiple scales - focusing solely on individual components or aggregate behavior provides an incomplete picture. The modular organization we observe at the macro level arises from complex interactions between specialized components rather than from components that are themselves modular.
>
>
> This also raises intriguing questions about how this functional organization is maintained. Do components develop fixed specialized roles during training, or is there dynamic coordination also at inference time where components adjust their behavior based on the outputs of previous components to ensure a reliable result? Understanding these mechanisms of self-organization could provide valuable insights into how robust high-level capabilities emerge from collections of specialized neural components.
>
>
> “The Hydra Effect: Emergent Self-repair in Language Model Computations” by McGrath et al. is an example of work in this direction, suggesting self-organisation at inference time.
>
> 4.
>
> > Your proof-of-concept for mitigating prompt injection shows promising results. Have you explored how this technique might scale to more diverse types of adversarial inputs or larger language models? What challenges do you anticipate in applying this method to real-world deployment scenarios?
>
> We haven't yet explored other settings, but here is how we envision the workflow could scale to real-world deployment scenarios:
> 1. **Identifying a context/request pattern in user prompt.** The method would apply to scenarios where a model executes a task on variable context data. This includes common use cases vulnerable to prompt injection like question-answering on webpages, document summarization, or translation services where malicious content could be embedded in the context.
> 2. **Identifying the request.** While our proof-of-concept focused on simple questions about cities, real-world applications would need to handle more complex requests like "translate this text" or "summarize these key points". The challenge is isolating the semantic meaning of these requests from their surface form.
> 3. **Dynamically create a trusted input.** For each request type, we would need to generate a trusted example by combining the identified request with verified safe context data. This provides a clean reference point for the request processing, free from potential prompt injections. The main challenge is automating this process while ensuring the trusted inputs stay representative of the user prompt.
> 4. **Multi-token request patching.** Our current method only handles single-token outputs, but real applications require generating multiple tokens. The key technical challenge is maintaining coherent request processing across multiple generation steps. This requires investigating whether the modular separation we observed extends to multi-token generation.
> We added this hypothetical workflow to Appendix C to help the reader understand how the preliminary results could apply to real-world cases.

---

> > ### Comment · Reviewer_diV7 · 2024-11-26
> > **Reviewer Response**
> >
> > Thanks for the clarifications that you have made. I appreciate it.
> > While it does clear up some of the concerns, I believe the paper will significantly benefit from a thorough revision.
> > For the time being, I have decided to keep my scores.
> > Thanks.

---

### Official Review · Reviewer_ha2n · 2024-10-31

**Soundness:** 3
**Presentation:** 3
**Contribution:** 3
**Rating:** 5
**Confidence:** 4

**Summary:**

The paper introduces ORION, a structured collection of retrieval tasks aimed at studying how language models (LMs) handle retrieval in various contexts. The research investigates 18 models, applying causal analysis to show that LMs use a modular approach to process retrieval tasks, with middle layers focusing on the request and late layers on retrieving context. The concept of request-patching is introduced to improve model oversight and counteract prompt injections.

**Strengths:**

- **Innovative Dataset**: ORION serves as a unique, structured dataset collection that enables comparative analysis across multiple models and retrieval scenarios.
- **Causal Analysis**: The study's use of causal interventions provides valuable insights into model behavior, highlighting a modular structure in how LMs process retrieval.
- **Proof of Concept**: Demonstrates a practical application for mitigating prompt injections using request-patching, enhancing model robustness in certain cases.
- **Thorough Evaluation**: The research assesses models across a range of sizes and tasks, reinforcing the applicability of findings.

**Weaknesses:**

- The methodology focuses heavily on retrieval tasks and may not be immediately applicable to more complex scenarios involving multi-step reasoning or decision-making.
- Although insightful, the microscopic component analysis indicates that variability in task-solving components remains complex and less well-explained by the macro-level findings.
- The proof of concept for oversight is limited to retrieval tasks with single attributes, raising questions about scalability to broader, real-world applications.
- Further exploration of how these findings apply to closed-source models or those with different architectures would strengthen the paper’s generalizability.

**Questions:**

1. Can the authors discuss potential extensions of ORION to handle multi-attribute or more complex reasoning tasks?
2. How might the request-patching approach be adapted or extended for use in real-world, multi-step tasks?
3. Did the analysis include any comparisons between models fine-tuned with instruction-following data and those that are not?
4. Are there potential limitations or risks in applying request-patching to models with different architectures or training methods?

---

> ### Author Response · Authors · 2024-11-17
>
> We thank the reviewers for their thoughtful comments.
>
> > 1. Can the authors discuss potential extensions of ORION to handle multi-attribute or more complex reasoning tasks?
> One straightforward extension to include multi-attribute question, would be to include prompts of the form:
> “Bob is living in Paris and 31 years old.
> Alice is 50 years old and lives in London.
> Question: what is the age of Bob?”
>
> One could use causal analysis to isolate which parts of the model are identifying the character associated with the question (‘Bob’), and the attribute ‘age’. Such work could also reveal that other attributes are used, for instance maybe the fact that ‘Bob’ appears first might turn out to be relevant.
>
> For more complex reasoning tasks, LLM typically solves those by using chain-of-thought prompting. An approach could be here to first decompose chain-of-thoughts into individual steps to include the crucial next-token prediction, for instance where the model predicts intermediate results.
>
> These crucial next-token predictions could then be analysed using causal analysis. One could imagine using an LLM to rewrite the partial chain of thoughts to key the global structure while modifying key variables. We would use the rewritten chain-of-thought as an input to patch from / to the original partial chain of thoughts.
>
> The challenge here would be i) identifying the ‘crucial’ next-token prediction, evaluating if one can focus on a few next-token prediction to understand the full chain of thoughts. and ii) the generation of counterfactual inputs that lead to meaningful patching experiment, for instance following a property similar to the ‘decomposable’ in ORION.
>
> > 2. How might the request-patching approach be adapted or extended for use in real-world, multi-step tasks?
>
> To include real-world tasks, we could study prompts of the form (context, instruction). Such as:
> “Here is a text: [Text].
> Instruction: Summarize / Translate in language X / Extract the key characters / Describe the sentiment of this text / etc.“
>
> As the model answer would be several token long, we could extend the request patching technique by repeatedly patching at every token for a fixed generation length, or until a stop token is predicted.
> This could evaluate if the request / context division holds true for these more realistic settings.
> For multi-step tasks, we could approach it using a similar strategy as the one discussed for chain-of-thought above.
>
> > 3. Did the analysis include any comparisons between models fine-tuned with instruction-following data and those that are not?
> Yes. We include falcon-7b, falcon-40b (base models), and falcon-7b-instruct, falcon-40b-instruct (the same model after fine-tuning with instruction-following data).
>
> The Figure 14 in Appendix D shows side-by-side results of request-patching at every layer, on a set of models including  these four models. The fine-tuned and non-fine-tuned demonstrate very similar results. This is coherent with the intuition that fine-tuning is only superficially affecting the model internals.
>
> > 4. Are there potential limitations or risks in applying request-patching to models with different architectures or training methods?
>
> We applied request-patching to a broad set of Transformer models, including variations in their architecture. The Pythia, Falcon and Llama 2 models use parallelized attention and MLP. Moreover, Falcon contains additional layer normalization at the input of modules. LLama 2 uses the SwiGLU activation function instead of  GELU, and layer normalization only at the input of modules.
>
> The experimental procedure and results are similar across these variations in architecture and training method (with the Falcon base and fine-tuned models). This suggests that this internal division of retrieval tasks is universal for Transformer langage models.
>
> One could expand the analysis to more exotic architecture such as state space models to check if this division holds for language models beyond Transformers.

---

### Official Review · Reviewer_7Spb · 2024-11-01

**Soundness:** 3
**Presentation:** 3
**Contribution:** 3
**Rating:** 6
**Confidence:** 4

**Summary:**

This paper introduces ORION, a diverse set of structured retrieval tasks designed to study how language models (LMs) handle complex information retrieval across six different domains. The authors conduct a comprehensive analysis on 18 open-source LMs with varying sizes and find that these models internally decompose retrieval tasks in a modular fashion. Specifically, middle layers process the request (e.g., a question), while later layers retrieve the relevant entity from the context. Through causal analysis, they demonstrate that enforcing this modular decomposition preserves a significant portion of the model's performance. Additionally, the paper presents a fine-grained case study and proposes a scalable method for internal oversight of LMs to mitigate prompt-injection attacks, significantly improving accuracy with minimal human supervision. This work provides insights into the universal modular processing of tasks in LMs and pioneers the application of interpretability for enhancing model oversight.

**Strengths:**

1. ORION represents a novel data-centric approach, facilitating a comparative study across 18 models and 6 diverse domains, enhancing the scope and depth of retrieval task analysis.
2. The fine-grained case study on Pythia-2.8b effectively bridges macroscopic and microscopic descriptions, adding a layer of detailed validation to the findings.
3. The insights gained from this study significantly contribute to the broader understanding of LM internals, influencing future design and optimization strategies.

**Weaknesses:**

1. Limited Generalizability Beyond ORION Tasks: While the study provides valuable insights into modular decomposition within the specific tasks included in ORION, there may be limitations in generalizing these findings to other types of tasks or real-world scenarios that are not represented in the ORION collection.

2. Dependency on Specific Models and Versions: The analysis is based on a specific set of 18 open-source language models with varying sizes, which may not fully represent the behavior of all existing LMs or future iterations of these models. Additionally, focusing on specific versions (e.g., Pythia-2.8b) might introduce limitations related to the unique characteristics or limitations of those versions.

**Questions:**

See the above weaknesses.

---

> ### Author Response · Authors · 2024-11-17
>
> Thank you for the review of our work. We’d like to contextualize the weaknesses mentioned to present how they reflect design choices that fit in a broader research program.
> 1.
> > Limited Generalizability Beyond ORION Tasks: While the study provides valuable insights into modular decomposition within the specific tasks included in ORION, there may be limitations in generalizing these findings to other types of tasks or real-world scenarios that are not represented in the ORION collection.
>
> We recognize the limitation of the tasks in ORION. We choose to focus on well-defined, simple retrieval tasks, as i) single token ii) can be executed by small models, expanding the range of models in our analysis and iii) it is easy to design retreival tasks following the same structure, but from different domains.
>
> We decided to start by grounding our work into diverse models and taks to establish broad experimental results, by ruling out dependency per task domain, model size, or architecture variation.
>
> From these initial results, we could expand the analysis towards more realistic tasks along different directions (copying from the answer to reviewer ha2n):
>
> **Multi-label.** One straightforward extension to include multi-attribute question, would be to include prompts of the form:
> “Bob is living in Paris and 31 years old.
> Alice is 50 years old and lives in London.
> Question: what is the age of Bob?”
>
> One could use causal analysis to isolate which parts of the model are identifying the character associated with the question (‘Bob’), and the attribute ‘age’. Such work could also reveal that other attributes are used, for instance maybe the fact that ‘Bob’ appears first might turn out to be relevant.
>
> **Chain of thought reasoning.** For more complex reasoning tasks, LLM typically solves those by using chain-of-thought prompting. An approach could be here to first decompose chain-of-thoughts into individual steps to include the crucial next-token prediction, for instance where the model predicts intermediate results.
>
> These crucial next-token predictions could then be analysed using causal analysis. One could imagine using an LLM to rewrite the partial chain of thoughts to key the global structure while modifying key variables. We would use the rewritten chain-of-thought as an input to patch from / to the original partial chain of thoughts.
>
> The challenge here would be i) identifying the ‘crucial’ next-token prediction, evaluating if one can focus on a few next-token prediciton to understand the full chain-of-thoguths. and ii) the generation of counterfactual inputs that lead to meaningful patching experiment, for instance following a property similar to the ‘decomposable’ in ORION.
>
> **Instruction in context.** We could study prompts of the form (context, instruction). Such as:
> “Here is a text: [Text].
> Instruction: Summarize / Translate in language X / Extract the key characters / Describe the sentiment of this text / etc.“
>
> As the model answer would be several token long, we could extend the request patching technique by repeatedly patching at every token for a fixed generation lenght, or until a stop token is predicted.
> This could evaluate if the request / context division holds true for these more realistic settings.
>
>
> 2.
>
> > Dependency on Specific Models and Versions: The analysis is based on a specific set of 18 open-source language models with varying sizes, which may not fully represent the behavior of all existing LMs or future iterations of these models. Additionally, focusing on specific versions (e.g., Pythia-2.8b) might introduce limitations related to the unique characteristics or limitations of those versions.
>
> The set of 18 models we chose include diversity across size, architecture variations, training method (base models and fine-tuned models) that covers a large part of the possible characteristics from available open-source langage models. The fact that the request-patching results hold across all these axes of variation is strong evidence that the phenomenon would generalize to models outside of the studied set, and future iterations.
>
> If the results don't hold on an unstudied model, this would be an interesting avenue of research to try to understand in which way this model deviates from the set of models we studied.
>
> The study on Pythia-2.8b can be seen as a ‘model organism’ in biology. We conducted an in-depth open-ended study to identify interesting phenomena before attempting to replicate it in other models. Our findings, even if only for a single model, are enough to challenge the intuitive view that the high-level mechanism would be directly translated at a low-level, opening the space of hypotheses one has to consider to make sense of this phenomenon.

---

> > ### Comment · Reviewer_7Spb · 2024-11-30
> > **Response to authors**
> >
> > Thank you for the author's response. Taking into account the comments of other reviewers and after comprehensive consideration, I decided to keep my score unchanged.

---

### Official Review · Reviewer_EUoL · 2024-11-04

**Soundness:** 4
**Presentation:** 4
**Contribution:** 4
**Rating:** 8
**Confidence:** 3

**Summary:**

This paper studies how LLMs internally solve retrieval tasks. To do this, they introduce ORION, a collection of 15 datasets and apply causal analysis to 18 LLMs. To solve the tasks in ORION, the LM has to combine information from the question and context into the last token position. The authors use residual stream patching on this last token to investigate how the LM decomposes and combines the information. They need 2 prompts for this; prompt1 = R1C1 and prompt2 = R2C2; R stands for Request (question) and C for context. They observe that depending on the layer where they do the patching, the model may output R1(C1) (answer to question 1 from context 1), R2(C2), or R1(C2). This means that the patching is intervening the model’s computation and combining high-level information from 2 prompts. With this, it is possible to supervise the internal process of the models and for example, make the model more robust to prompt injection.

**Strengths:**

* Extensive experiments on 18 LMs from 125M to 70B parameters on 15 datasets.
* Empirical evidence of internal decomposition of retrieval tasks. This can be very useful of the community to better understand the internal behavior of the models and improve their interpretability.
* Application to mitigate prompt injection

**Weaknesses:**

* It remains unclear to me how to operationalize the request patching to mitigate prompt injection. It seems it requires a trusted input although lines 420 seem to indicate it doesn't. See more details on the question below.

**Questions:**

* in lines 450-451, it is written that you perform residual stream patching from the trusted input to an untrusted input. However, I couldn't understand how you obtain the trusted input. My understanding is that we only have an untrusted input in a real-world scenario, so this method is difficult to use. Line 420 says that you designed the application so that it doesn't require access to ground-truth labels, but isn't the trusted input the ground-truth label? To organize my questions a bit more:
   * What is this "trusted input" and how it differs from the ground-truth input?
   * How do you obtain the "trusted input"?

---

> ### Author Response · Authors · 2024-11-17
>
> Thank you for the review of our work!
>
> 1.
> > What is this "trusted input" and how it differs from the ground-truth input?
>
> On line 420 we claim that “we design our application so that it does not require access to ground-truth labels”, we don’t mention ground truth *input*. This might be the source of the confusion.
>
> The broader motivation for this proof of concept is the case where human supervisors cannot evaluate the output of a model. In other words, we ask a question to the model, we get an answer, but we cannot check if this answer is correct or not, i.e. we cannot produce ground-truth labels.
>
> However, the supervisors can make the difference between ‘trusted’ context, eg. carefully curated data during the development phase, and ‘untrusted’ context, eg. during the deployment phase, where the inputs could be subjected to attack such as prompt-injection.
>
> This proof of concept shows that even in this case, we can ensure more robustness in the untrusted context by patching one piece of the mechanism from the trusted context to the untrusted context.
>
> Of course, in our case, the task is simple and we have access to the correct answer, this is what we use to measure the accuracy in Table 1. However, these labels are not needed to perform request-patching.
>
> Thanks to your comment, we clarified the line 420 in the paper.
>
> 2.
> > How do you obtain the "trusted input"?
>
> The “trusted input” is the concatenation of a randomly picked story and a question (“what is the city?”). We call it “trusted” because it is a context where the model is operating in normal condition.
>
> Let us know if this doesn't answer your question.

---

> > ### Comment · Reviewer_EUoL · 2024-11-25
> >
> > Thank you for your answer. Yes, it has answered my questions.
> >
> > I still believe the application of this is limited to very specific scenarios with somewhat predefined inputs and vulnerable to prompt injections. Is my understanding correct? (regardless of the application, the knowledge about the internal workings of the LLM learned from this paper is valuable IMHO)

---

> > > ### Author Response · Authors · 2024-11-26
> > >
> > > Thank you for your answer. This is right that the current application can only deal with very restricted scenarios.
> > >
> > > To make it easier to inspire future applications in realistic setting, we outlined a plan, and key challenges to overcome to build from this toy example to real-world conditions (copying from the response to Reviewer diV7, we also added it to Appendix D). This might be of interest to you to evaluate the impact of the proof of concept.
> > >
> > > 1. **Identifying a context/request pattern in user prompt.** The method would apply to scenarios where a model executes a task on variable context data. This includes common use cases vulnerable to prompt injection like question-answering on webpages, document summarization, or translation services where malicious content could be embedded in the context.
> > > 2. **Identifying the request.** While our proof-of-concept focused on simple questions about cities, real-world applications would need to handle more complex requests like "translate this text" or "summarize these key points". The challenge is isolating the semantic meaning of these requests from their surface form.
> > > 3. **Dynamically create a trusted input.** For each request type, we would need to generate a trusted example by combining the identified request with verified safe context data. This provides a clean reference point for the request processing, free from potential prompt injections. The main challenge is automating this process while ensuring the trusted inputs stay representative of the user prompt.
> > > 4. **Multi-token request patching.** Our current method only handles single-token outputs, but real applications require generating multiple tokens. The key technical challenge is maintaining coherent request processing across multiple generation steps. This requires investigating whether the modular separation we observed extends to multi-token generation.

---

### Meta-Review · Area_Chair_VRfp · 2024-12-21

**Metareview:**

This paper studies how language models internally perform a retrieval task – identifying relevant information from a lengthy, complex input text. To study this, the paper first introduces a new dataset called ORION, consisting of such tasks in several different scenarios. Then, the paper evaluates 18 open-source LMs, and finds that the model internally decomposes the retrieval task into two stages — processing the request and retrieving the correct entity — through intermediate layers of Transformers. The paper then apply this finding to a proof of concept for mitigating prompt injection.

Strengths
- Experiments are comprehensive, consisting of 18 LMs with up to 70B parameters on 15 tasks (EUoL, 7Spb, ha2n, diV7).
- Empirical evidence of internal decomposition of the retrieval task is convincing (EUoL, ha2n, diV7).
- The detailed analysis with Pythia 2.8B is valuable (7Spb).
- Proof of concept for prompt injecting using request patching is valuable (ha2n, EUoL).

Weaknesses
- Proof-of-concept for mitigating prompt injections remains limited in scope (7Spb, EUoL, diV7), i.e., how to operationalize the request patching to mitigate prompt injection.
- Whether the findings generalize beyond ORION is unclear, e.g., the data mainly confuses on single-attribute requests, and how this generalizes to more complex retrieval scenarios involving multiple attributes or nested queries remain underexplored (7Spb, diV7, ha2n).
- Inconsistency in macro-level and micro-level modularity remains unresolved (diV7).
- Potential limitation of the residual patching technique is not discussed (diV7).

**Additional Comments On Reviewer Discussion:**

Author responses mainly responded to questions, and reviewers kept the scores.

---

### Decision · Program_Chairs · 2025-01-22

Accept (Poster)